# Characteristics of Resuspended Road Dust with Traffic and Atmospheric Environment in South Korea

Sungjin Hong , Hojun Yoo , Jeongyeon Cho , Gyumin Yeon and Intai Kim *

Department of Transportation Engineering, Myongji University, Youngin 17058, Korea; ahsj9927@mju.ac.kr (S.H.); dbghwns0@naver.com (H.Y.); wjddus942@naver.com (J.C.); bx1002@naver.com (G.Y.)
* Correspondence: kit1998@mju.ac.kr; Tel.: +82-10-9767-9017

**Abstract:** Characterizing the influencing factors of resuspended dust on paved roads according to the atmospheric environment and traffic conditions is important to provide a basis for road atmospheric pollution control measures suitable for various road environments in the future. This study attempts to identify factors in the concentration of resuspended dust according to the level of road dust loading and $PM_{10}$ emission characteristics according to atmospheric weather environment and traffic conditions using real-time vehicle-based resuspended $PM_{10}$ concentration measuring equipment. This study mainly focuses on the following main topics: (1) the increased level of resuspended dust according to vehicle speed and silt loading (sL) level; (2) difference between atmospheric pollution at adjacent monitoring station concentration and background concentration levels on roads due to atmospheric weather changes; (3) the correlation between traffic and weather factors with resuspended dust levels; (4) the evaluation of resuspended dust levels by road section. Based on the results, the necessity of research to more appropriately set the focus of analysis in order to characterize the resuspended dust according to changes in the traffic and weather environment in urban areas is presented.

**Keywords:** silt loading; resuspended dust and lane traffic level; vehicle-based real-time PM measured system; yellow dust; $PM_{10}$

## 1. Introduction

Particulate matter (PM) with diameters of less than 10 μm can cause various damages to the human body, such as respiratory failure and mental illness [1]. Additionally, their concentrations are often elevated close to roadways [2]. PM is caused through the contribution of effective road management, so the degree of contamination must be determined by including a quantitative analysis of contamination to determine if it is important enough to implement control measures. Road dust (RD) is generated and diffused by traffic, and it is one of the main causes of road pollution in cities. It is categorized as an exhaust emission (EE). For example, exhaust emissions include emissions from internal combustion locomotives and automobiles. In addition, nonexhaust emissions (NEEs) are generated by vehicles driving on roads [3]. In recent decades, it has been known that direct emissions from tailpipes have been reduced significantly due to the strengthening of exhaust emission standards applied to vehicle air masses and the extensive implementation of catalytic converters and diesel particulate filters [4]. However, it has been suggested that the main cause of failure to derive an effective reduction in $PM_{10}$ in large cities is the undervaluation and neglect of NEEs, which consist of road dust, tire wear, road wear, and brake wear particles caused by tire–road friction [5]. It also emphasizes that emissions from road traffic contribute to PM concentrations to at least the same extent as exhaust emissions [3,4,6–8].

According to the 2017 Clean Air Policy Support System (CAPSS), which provides the extent of the emissions of air pollutants in the Republic of Korea, resuspended dust from roadways makes up 45% of the total concentration of dust. Therefore, as an alternative to the main management of suspended dust in urban components, road cleaning

programs by local governments are used extensively. It is important to properly design and implement road cleaning in consideration of road dust characteristics and climate and weather characteristics because of the economic uncertainty caused by short-term effects without considering cleaning methods, regional characteristics, and weather characteristics [9]. However, the reason for the difficulty of establishing the characteristics of dust that is resuspended from roadways is that the study to estimate the concentration of resuspended dust through the establishment of related processes and modeling is still at the basic level [10,11]. Furthermore, due to the large fluidity characteristics of very small substances, i.e., smaller than $PM_{10}$, research to determine the chemical profile of dust on roads composed of meteorological environments, transportation activities, and construction requires significant resources and time.

In this study, the concentration of silt loading (sL), resuspended dust of $PM_{10}$, and floating dust $PM_{10}$ on paved roads separated by space was measured, and the results were used to analyze the concentration of resuspended dust on paved roads in urban areas. The $PM_{10}$ concentration measured during operation quantified the characteristics of $PM_{10}$ based on the analysis of $PM_{10}$ information of the air pollution monitoring station, weather information of the competent meteorological station, and traffic information measured through the use of a vehicle detection system (VDS). Previous studies have shown the physical and chemical properties of fine dust on Mars by testing re-entrained aerosol kinetic emissions from road (TRAKER) equipment for real-time monitoring. The equipment has the advantage of being able to immediately investigate dust emissions on many roads with long sections [12–14]. This clearly states that the fine dust concentration measured by the system can be used to quantify the relationship between major factors for increasing road particle emissions by utilizing repeated measurements of resuspended dust to supplement the realistic limitations of AP-42. Various studies in these fields were mainly for identifying the emission characteristics or chemical profiles of the resuspended dust from roads [15–21] and a few studies considered the spatial and time variations in the major parameters [22].

The main purpose of this study was to provide an experimental basis for future road environmental changes by analyzing and characterizing the main emission characteristics of the $PM_{10}$ concentrations measured on paved roads that are caused by sL, air pollution, meteorological variables, and traffic conditions. In this study, we focused on this and tried to establish a process to secure data reliability by constructing a resuspended $PM_{10}$ measurements system and by observing the concentration of resuspended dust in separate sections of a road, i.e., sections less than 500 m in length.

## 2. Materials and Methods

### 2.1. Study Area

To present a reasonable classification of the characteristics of $PM_{10}$ emissions on roads, locations were selected that were representative of vehicle-operating conditions and environmental characteristics. Measurements were held during periods without traffic impact observed in four sections from September to October 2021. During the measurement period, there were no weather events such as rainfall (0.01 inches of rain) that could have had a major effect on the fine dust concentration [23,24]. In addition, another measurement was determined on the concentration of resuspended dust on paved roads where traffic operated (Figure 1). Additionally, background $PM_{10}$ data (Bg) and weather variables and traffic condition data were collected through the urban atmosphere and traffic monitoring system at the same time. The traffic data were collected in six directions in three adjacent areas where traffic volume VDS was installed (Table 1).

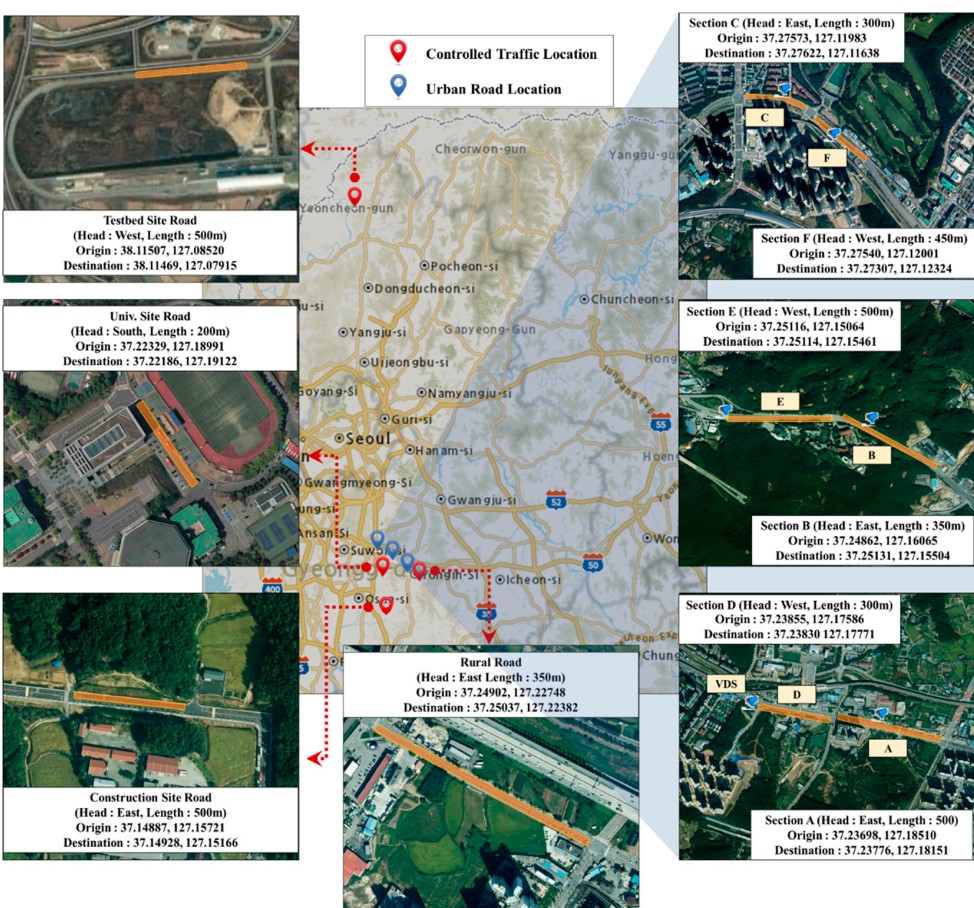

**Figure 1.** Study area in South Korea (Controlled traffic location and urban road location).

**Table 1.** Road conditions of the measurement section and distance from the observation.

| Category | Site | Length (m) | Road Width/Condition | Distance of the Observation | |
|---|---|---|---|---|---|
| | | | | City Bg PM$_{10}$ | Meteorological Variables |
| Controlled Traffic | Testbed | 500 | One-lane road | With a radius of 2.5 km | With a radius of 24 km |
| | Construction Site | 500 | Two-lane road | With a radius of 3.5 km | With a radius of 20 km |
| | Univ. | 200 | One-lane road | With a radius of 1.6 km | With a radius of 14.5 km |
| | Rural | 350 | Two-lane road | With a radius of 2.8 km | With a radius of 17.5 km |
| Operating road | A | 500 | Four-lane road/ 905 (Veh/time/day) | With a radius of 8 km | With a radius of 14.5 km |
| | B | 350 | Three-lane road/ 633 (Veh/time/day) | With a radius of 5 km | With a radius of 11.5 km |
| | C | 300 | Four-lane road/ 1006 (Veh/time/day) | With a radius of 1 km | With a radius of 8.2 km |
| | D | 300 | Four-lane road/ 850 (Veh/time/day) | With a radius of 8 km | With a radius of 13.2 km |
| | E | 500 | Three-lane road/ 431 (Veh/time/day) | With a radius of 5 km | With a radius of 11 km |
| | F | 450 | Four-lane road/ 697 (Veh/time/day) | With a radius of 1 km | With a radius of 8.5 km |

### 2.2. Resuspended PM$_{10}$ Measuring System

Road dust generated by vehicles driving on urban roads is known to have the greatest impact on the concentration of PM depending on the size of the sL collected on the surface of the road and the load and speed of the vehicles being driven on the road [11,14]. However, the method of measuring fine dust by performing and comparing direct dust collection activities through human activities on roads has safety and efficiency problems, and there

are practical limitations in observing real-time concentrations. Mobile resuspended dust measurement systems were developed and implemented in the early 2000s. Scamper, TRAKER, and others generally use a method to calculate the difference between background concentration and tire measurements [12].

By establishing a real-time mobile dust measurement system based on the Desert Research Institute (DRI)'s TRAKER method, the temporal and spatial distribution characteristics of the resuspended dust on urban pavement can be checked and the impact of road dust sources can be confirmed by traffic and environmental factors [25].

The resuspended $PM_{10}$ measurement system (Figure 2) used an optical sensor that inhaled PM every second to determine the $PM_{10}$ concentration on the road through the front (mounted on the hood) inlet of the vehicle and the rear (right front wheel) inlet of the tire [25]. In quantifying the optical and physical characteristics of resuspended dust, the measurement of real-time dust concentration used scattered light to measure the weight of dust passing through. The measurement method clearly states that the concentration of resuspended dust can be measured [26].

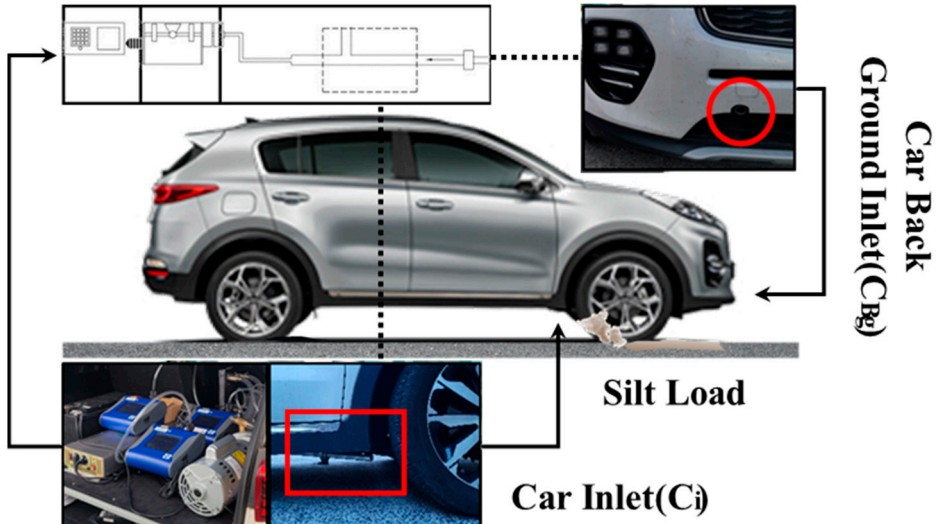

**Figure 2.** Configuration of the resuspended $PM_{10}$ measurement system.

To summarize, the equipment contains six main components, i.e.,:

1.  Vehicle specifications: 2016 SUV Sportage 2.02 WD, Tire Model 225/60R17.
2.  $PM_{10}$ Sensor: The 8530 Dust-Trak Optical PM Sensor Model (TSI Company, Shoreview, MN, USA) with $PM_{10}$ inlet is a portable device that can measure the concentrations of various sizes of dust particles in real time, such as $PM_{10}$ and $PM_{2.5}$.
3.  Inlet: The right tire line inlet is 175 mm above the ground and 50 mm behind the tire. The $PM_{10}$ concentration ($C_i$) obtained from the back of the tire during movement indicates that the air velocity measured along the center of the tire was not affected significantly by the surrounding wind direction at a distance less than 100 mm from the tire [27].
4.  Uniform Velocity Flow Sampling Inhalation: A vacuum pump is controlled by a customized suction port to adjust the pressure of the suction port appropriately, thereby producing a pressure-free suction port, so the air flow that is being sampled has the same fluid flow as the surrounding air flow.
5.  Data Collection System: Laptop computers were used to collect and transmit Dust-Trak data at one-second intervals.
6.  Monitoring Camera: A camera capable of monitoring the inside and outside of the vehicle can be installed to visually cross-verify the measurement data in the driving section and monitor the speed observed on the vehicle's speedometer.

### *2.3. AP-42 Silt Loading Collection*

In order to collect fine dust on the surface of a paved road, a section of the road was sampled using an 800 W vacuum cleaner, 3 m$^2$ frame of silt loading sampling, a fine dust brush, and a 3 kW gasoline generator [23].

Dust was immediately collected in the chamber, and it was eventually collected from the quartz fiber filters. At least three samples with different aerodynamic diameters were taken from a surface area of 3 m$^2$ for field sampling to reduce errors [28]. The samples were weighed using 200 mesh, standard Taylor screens, and a sieve filter machine.

### *2.4. Collecting City PM$_{10}$, Meteorological Variables, and Traffic Data*

The data were collected as external impact indicators to analyze the effects of changes in the meteorological variables and traffic conditions on the PM$_{10}$ concentration levels observed through the resuspended PM$_{10}$ measurements system [29,30]. The collected elements were PM$_{10}$, the temperature of the ground (Gr temp), relative humidity (RH), wind speed (WS), sea level pressure, visibility, and time of after rainfall (rainfall time). The traffic data observed with adjacent VDS devices were divided into sections during the measurement period.

### 3. Results and Discussion

### *3.1. Verification of the Real-Time Measuring System*

As a method of quantifying the concentration of road dust by sections using a mobile, real-time, PM$_{10}$ monitoring system, the concentration data measured with the inlet built into the front bumper of the vehicle were set to the background PM$_{10}$ concentration (C$_{Bg}$). This value was obtained by subtracting the background concentration from the resuspended PM$_{10}$ concentration (C$_i$) measured using the dust resuspended on the road along with the background concentration on the road through friction between the tires and the surface of the road. This value was quantified [31]. C$_{res}$ was calculated according to the following equation:

$$C_{res} = \frac{1}{n} \cdot \sum_{i=1}^{n} C_i - C_{Bg} \tag{1}$$

where C$_{res}$ is the concentration of resuspended dust from the road surface ($\mu g/m^3$), C$_i$ is the PM$_{10}$ concentration measured in inches on the back of the tire resuspended dust concentration ($\mu g/m^3$), C$_{Bg}$ is the PM$_{10}$ concentration measured at the front bumper inlet of the vehicle background concentration ($\mu g/m^3$).

Monitoring was performed to analyze the change in the concentration of resuspended dust due to sL in the space and time, where the concentration measured during the interval transit time varied by the inlet. The large changes meant that the C$_{Bg}$ measurement was more than 100 $\mu g/m^3$ within a short measurement period, and *C$_i$* measured on the back of the tire was more than 1000 $\mu g/m^3$. Unlike the sL effects, this was proven to be an effect of the road dust as well as the external effects, such as the exhaust gas of other vehicles and the addition and deceleration of vehicles. Considering the average PM$_{10}$ concentration was 50 $\mu g/m^3$ and the 24 h average PM$_{10}$ concentration was 100 $\mu g/m^3$ level, it was excluded as an outlier. This was based on previous studies that indicated that the change in the C$_{Bg}$ level when the vehicle was not moving was not affected significantly by the exhaust of other vehicles, and the concentration of C$_i$ measured on the back of the tire could be used as the concentration of resuspended dust [13].

The process of verifying that the monitoring system can perform data collection functions for quantifying the concentration of road dust was conducted under three experimental conditions that could determine whether the measured concentration of the dust was significant when the vehicle moved.

(1) We ensured that the measured concentration did not vary significantly when the vehicle stopped, and tried to calculate the range of the measured speed by comparing

the concentration measured by increasing the speed as it passed the section at a constant speed.

(2)    If a certain measurement speed was maintained, the average $C_{res}$ concentration measured in a short section and the amount of dust accumulated on the surface of the road and the concentration of the resuspended dust due to vehicle travel were quantified by the sL collected at three different locations in the section.

(3)    The correlation was analyzed by comparing the concentration of $C_{res}$ measured by the size of sL in different sections (regions) selected by classification according to the major causes of the inflow of the expected material.

In the absence of peripheral vehicles, the monitoring system maintained a constant stop period and a constant speed, and the $C_{Bg}$ value measured during the driving period maintained a constant slight change of approximately 2 to 6 µg/m$^3$ per second. When the sL level collected in the section was 0.43 g/m$^2$ for the mean and 0.16 g/m$^2$ for the SD, the concentration was measured by maintaining 30, 40, 50, and 60 km/h, respectively, to analyze the effect on changes in the $C_{res}$ concentration as the speed increased during the section (Table 2).

**Table 2.** $C_{res}$ concentration mean and standard deviation value measured with car speed in testbed.

| Car Speed (km/h) | $C_{res}$ Mean (µg/m$^3$) | $C_{res}$ SD (µg/m$^3$) |
|---|---|---|
| 30 | 23.0 | 3.4 |
| 40 | 54.6 | 14.8 |
| 50 | 69.1 | 18.9 |
| 60 | 93.0 | 21.3 |

The expression for increasing $C_{res}$ concentration by velocity was the same as in Equation (2) (Figure 3).

$$y = 7.2971e^{0.0442x} \tag{2}$$

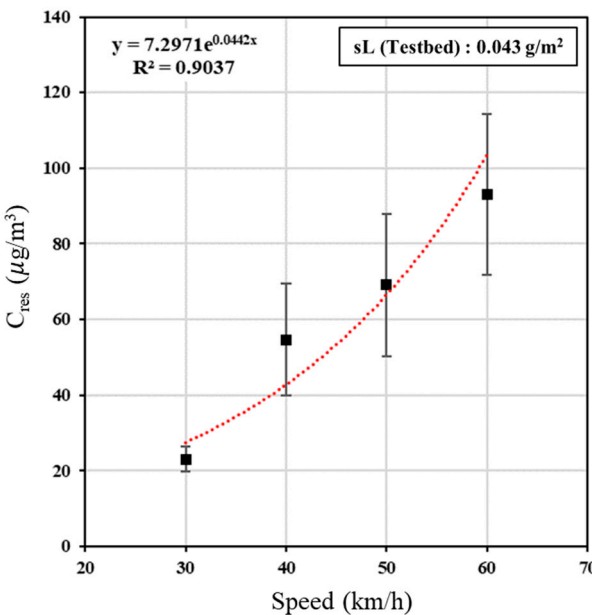

**Figure 3.** Increased $C_{res}$ due to increased driving speed at sL (0.043 g/m$^2$).

### 3.2. sL and Resuspended Dust on Paved Roads with Different Inflow Conditions

The results of measuring the concentration of sL and resuspended dust in sections with different fine dust generation and management environments while maintaining a speed

of 50 km/h, which usually is used as the speed limit on roads operating in the center of the city, are presented in Table 3 and Figure 4. The sL collected at the entrance and exit of the construction vehicle next to the cement plant, where the traffic of medium-sized vehicles mainly occurred, was the highest at 0.093 g/m$^2$, and the concentration of resuspended dust was observed to be 225 μg/m$^3$. For four consecutive days on the controlled testbed and low-traffic, university roads, the sL levels measured at the same time were found to continue to increase in deposition in the absence of specific events for rainfall, and the weather conditions observed were at a 11 μg/m$^3$ $C_{res}$ concentration (Table 3). As an influencing factor of resuspended dust generated on paved roads, it is known that the concentration level deposited on the section varies greatly depending on the heavy vehicle configuration and characteristics of traffic [32]. It has been shown that the resuspended dust from the road due to vehicle travel can be deposited, removed, moved into the air [23], or affected by weather conditions, geometric conditions, building geometry, etc. [29,33,34], or by regular road cleaning [9].

**Table 3.** Concentration of sL, resuspended PM$_{10}$ on paved roads without any traffic.

| Section | | sL 1 (g/m$^2$) | sL 2 (g/m$^2$) | sL 3 (g/m$^2$) | Mean sL (g/m$^2$) | Bg PM$_{10}$ (μg/m$^3$) | Mean $C_i$ PM$_{10}$ (μg/m$^3$) | Mean $C_{Bg}$ PM$_{10}$ (μg/m$^3$) | Mean $C_{res}$ PM$_{10}$ (μg/m$^3$) |
|---|---|---|---|---|---|---|---|---|---|
| Construction Site Road | | 0.115 | 0.089 | 0.074 | 0.093 | 30 | 242 | 17 | 225 |
| Testbed Road | Day 1 | 0.024 | 0.038 | 0.039 | 0.034 | 13 | 51 | 4 | 47 |
| | Day 2 | 0.043 | 0.044 | 0.042 | 0.043 | 18 | 77 | 4 | 73 |
| | Day 3 | 0.062 | 0.068 | 0.063 | 0.064 | 26 | 93 | 11 | 81 |
| | Day 4 | 0.069 | 0.08 | 0.075 | 0.075 | 21 | 91 | 10 | 81 |
| Univ. Site Road | Day 1 | 0.002 | 0.002 | 0.002 | 0.002 | 10 | 40 | 11 | 29 |
| | Day 2 | 0.003 | 0.003 | 0.003 | 0.003 | 16 | 41 | 17 | 24 |
| | Day 3 | 0.005 | 0.005 | 0.007 | 0.006 | 23 | 50 | 22 | 28 |
| | Day 4 | 0.009 | 0.01 | 0.008 | 0.009 | 36 | 58 | 37 | 21 |
| Rural Road | | 0.007 | 0.006 | 0.003 | 0.005 | 58 | 77 | 66 | 11 |

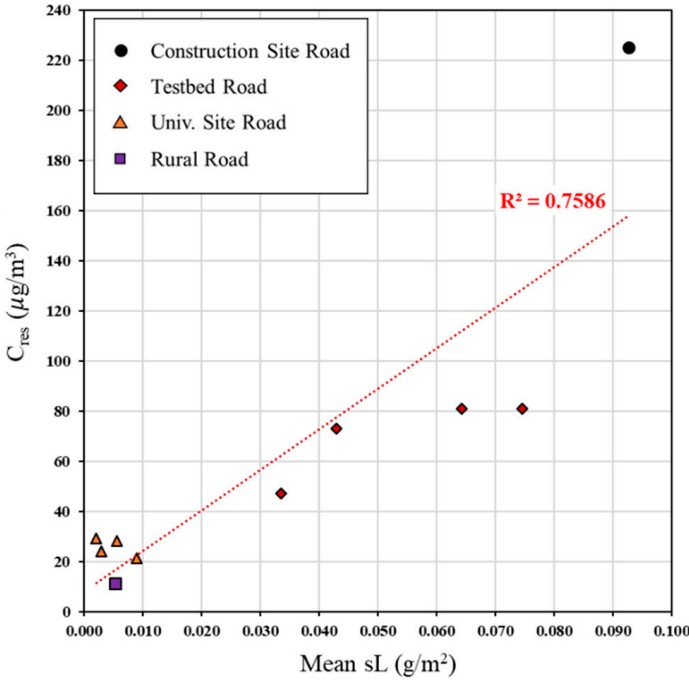

**Figure 4.** Trend of $C_{res}$ growth due to sL level in other measured sites.

An analysis of the correlation between the $C_{res}$ concentration and the collected sL measured when no ambient traffic occurred during the measurement and the space maintained was constant at 50 km/h speed on other roads showed a strong correlation, i.e., $R^2 = 0.7586$ (Figure 4). Depending on the increase in the sL level, the amount of $C_{res}$ increases depended on the site being measured, which could vary depending on the source of the dust, but roughly represented a pattern in which the amount of resuspended dust along with the sL increased with the size of the sL.

### 3.3. Relationship between Section Background Concentration of Dust and Its Resuspended PM$_{10}$

Due to the relationship between $C_{Bg}$ measured through the monitoring system and dust $C_i$ resuspended due to driving, the impact of the sL size varied from site to site (Figure 5). If the observed concentration level was closer to the Y-axis, the tendency to increase the concentration of $C_i$ compared to $C_{Bg}$ was stronger, and the size of sL was less than 0.002 to 0.009 g/m$^2$ compared to 0.034 to 0.093 g/m$^2$, and the inclination of the $C_{Bg}$ and $C_i$ concentrations were closer to one. When the size of the sL gradually became higher than 0.034 g/m$^2$, it was observed that the concentration level of $C_i$ increased linearly to show a difference of more than 47 to 225 µg/m$^3$. In the 0.002 to 0.009 g/m$^2$ range, the difference of the $C_i$ level of $C_{Bg}$ was 11 to 29 µg/m$^3$.

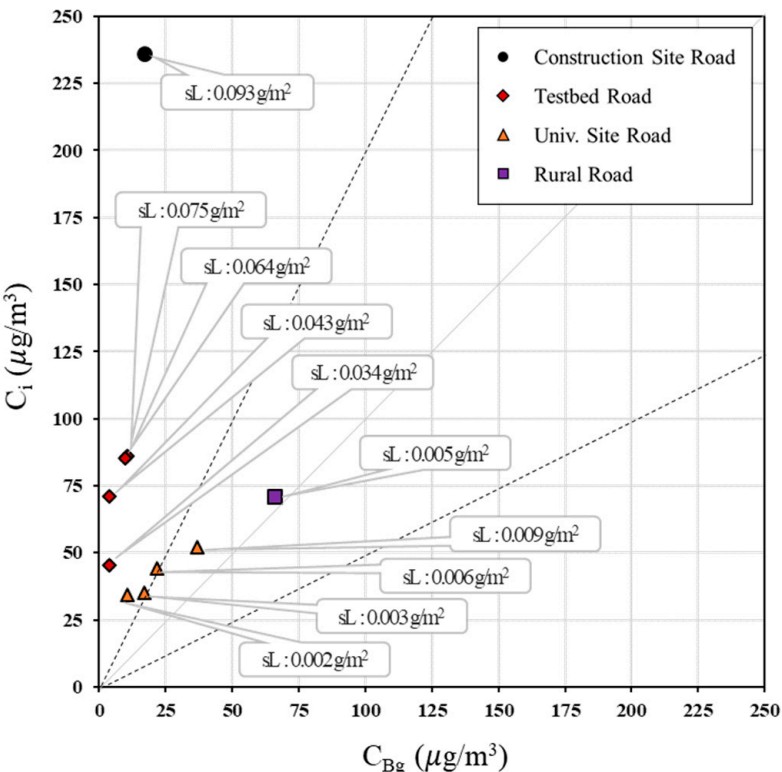

**Figure 5.** $C_{Bg}$ vs. $C_i$ concentration with sL in other measured sites.

Such a level of sL below 0.01 g/m$^2$ was a level that could be collected on roads where traffic was continuous, such as general highways and highways where particulate matter was not special [35] and low resuspended dust was observed. $C_{Bg}$ measured in a surrounding vehicle-free environment was found to increase with the level of air pollution (Bg) measured at adjacent stations at each measurement site, but the correlations were analyzed and found to have a strong correlation of $R^2 = 0.8484$ (Figure 6).

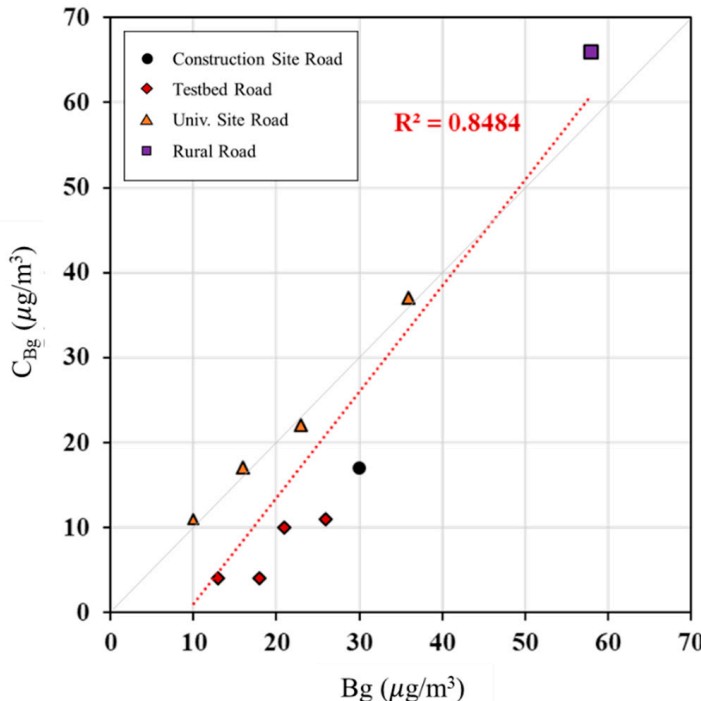

**Figure 6.** Compared concentration of Bg and $C_{Bg}$ on paved roads (without any traffic).

The measured $C_{Bg}$ was 4 to 14 μg/m$^3$, in Bg 10 to 20 μg/m$^3$, 10 to 22 μg/m$^3$, in Bg 20 to 30 μg/m$^3$, 17 to 37 μg/m$^3$, in Bg 30 to 40 μg/m$^3$, and 50 to 60 μg/m$^3$ in Bg 50 to 60 μg/m$^3$. However, the increase may have varied depending on the spatial difference between the observed sections. This level was considered to be a significant result when comparing the report [36,37] that the mortality rates of all diseases, cardiopulmonary disease, and lung cancer increased significantly by 4%, 6%, and 8%, respectively, for every 10 μg/m$^3$ increase in the concentration of particulate matter.

The relationship between Bg and $C_{Bg}$, measured during the running of the monitoring system on urban roads, was divided into two main patterns, i.e., (1) the yellow dust storm effect and (2) the no yellow dust storm effect. The yellow dust storm effect is a kind of dust fog, which is a phenomenon in which a large amount of yellow dust, which is called the continental yellow dust, drifts over a given area [38], indicating that the level of Bg and $C_{Bg}$ measured at adjacent stations can vary greatly. The correlation between Bg and $C_{Bg}$ measured on roads during traffic operation was relatively low compared to that of Bg and $C_{Bg}$ measured on roads with limited traffic, and $R^2 = 0.4697$ was shown when the yellow dust phenomenon was excluded (Figure 7). It was found that differences in $C_{Bg}$ also existed between sections in the same Bg and weather conditions observed by time zones at adjacent measuring stations. The difference was 20.45 μg/m$^3$, with different levels of $C_{Bg}$ on roads where the main sources of fine dust and inflow were judged to be similar. In addition, the occurrence of yellow dust was characterized by the opposite characteristics of the correlation $R^2 = 0.1033$ as the Bg level of the adjacent station became very high. The Regional Bg could be expected to have a quantitative impact on the overall pollution on roads, but when yellow dust occurred, the difference between $C_{Bg}$ and Bg on roads could cause different concentration levels of $C_{res}$ in the process of the continuous floating of road resuspended dust.

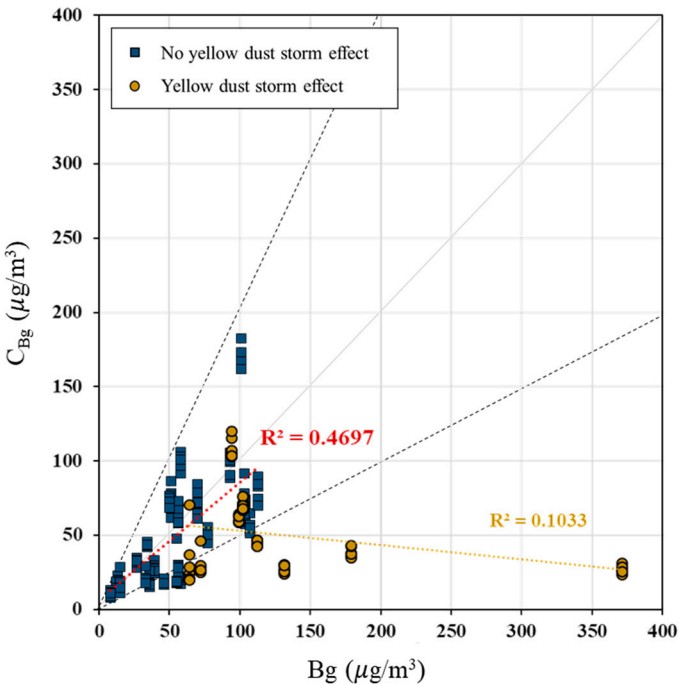

**Figure 7.** Concentration of Bg vs. $C_{Bg}$ on paved roads during traffic operation (classification due to the effects of yellow dust).

The correlation between $C_{Bg}$ and $C_i$ concentrations measured on roads during traffic operation showed a strong correlation of $R^2 = 9253$ without the influence of yellow dust, a weak correlation of $R^2 = 0.4256$ with yellow dust, and a correlation of $R^2 = 0.8031$ in respect to the overall concentration (Figure 8). The average concentration at which $C_i$ was derived was 226 μg/m$^3$, and the $C_i$ concentration measured at the rear of the tire was the highest compared to $C_{Bg}$.

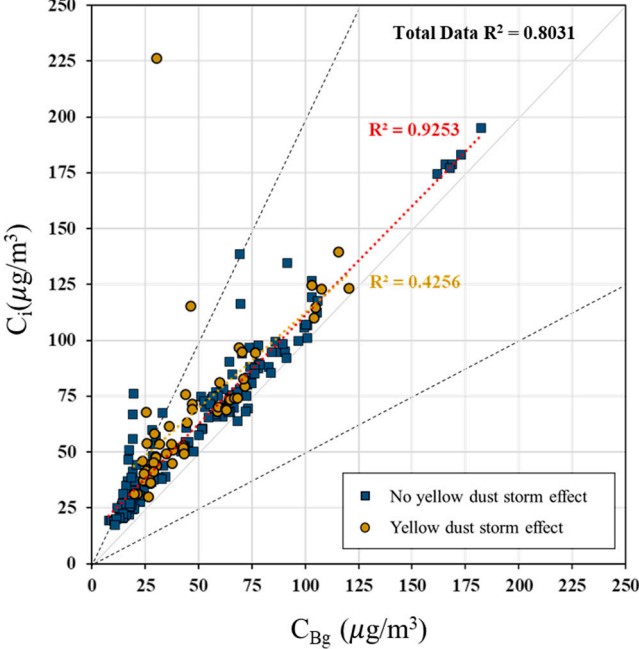

**Figure 8.** Concentration of $C_{Bg}$ vs. $C_i$ on paved roads during traffic operation (classification due to yellow dust impact).

### 3.4. Analysis of PM$_{10}$ Concentration Characteristics of Resuspended Dust by Influencing Factors

For the correlation analysis between the concentration of resuspended dust measured on the road, atmosphere, meteorological variables, and traffic factors, dust measured data were collected (excluding yellow dust days data from the entire data) in six sections.

To analyze the effects of the average C$_{res}$ concentration observed by road section on atmospheric and meteorological environments and traffic conditions, the C$_{res}$ data samples were divided into five grades within the statistical analysis. (Lower quartile (25th): 7.7 µg/m$^3$; median: 12.7 µg/m$^3$; upper quartile (75th): 15.6 µg/m$^3$; high C$_{res}$ (sL level: over 0.009 g/m$^2$): 29 µg/m$^3$ were used.)

- Low-level type: under 7.7 µg/m$^3$.
- Low- to middle-level type: 7.7 to 12.7 µg/m$^3$.
- Middle- to high-level type: 12.7 to 15.6 µg/m$^3$.
- High-level type: 15.6 to 29 µg/m$^3$.
- Bad-level type: over 29 µg/m$^3$ and the number of measurements by C$_{res}$ grade was the same as Figure 9.

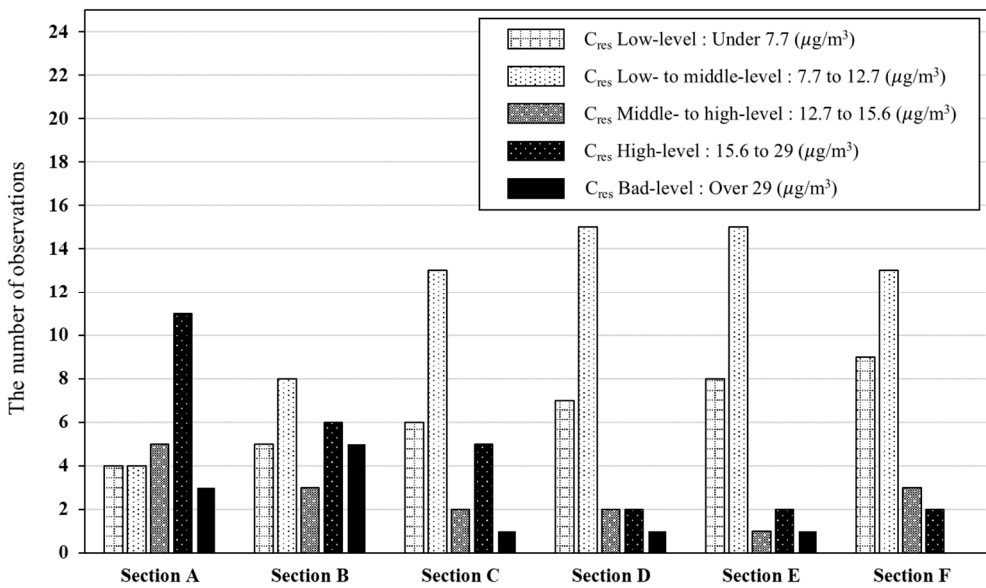

**Figure 9.** Comparison of the number of C$_{res}$ level type observations.

Among them, the results of C$_{res}$ measured as the bad level by section were observed in a wide range of Bg up to 15 to 113 µg/m$^3$ (Table 4). In different measured sections, C$_{res}$ with the bad level was observed even at 15 µg/m$^3$ in the level of concentration of Bg, and did not increase linearly. It is known that the Bg concentration and weather effects have a major influence on road atmosphere pollution, but it is difficult to represent them thoroughly, and it can be expected that external factors that are applied differently for each section are measured in the same time.

Since PM$_{10}$ on roads is known to be generated and diffused through various traffic elements, including traffic volume, traffic speed, road environment, and virtual elements [33], it is important to quantify the effect on the PM$_{10}$ concentration by reflecting the traffic characteristics of each section. The order of average C$_{res}$ concentration and traffic by liver level was Location 2—Section B (18.0 µg/m$^3$, 296 veh/time/lane) > Location 1—Section A (17.1 µg/m$^3$, 324 veh/time/lane) > Location 3—Section C (12.7 µg/m$^3$, 341 veh/time/lane) > Location 1—Section D (10.0 µg/m$^3$, 318 veh/time/lane) > Location 2—Section E (9.4 µg/m$^3$, 212 veh/time/lane) > Location 3—Section F (8.5 µg/m$^3$, 245 veh/time/lane) (Figure 10).

**Table 4.** Comparison of observed $C_{res}$ bad level concentrations by section and Bg concentration of the same time zone.

| Section | $C_{res}$ ($\mu g/m^3$) | Bg ($\mu g/m^3$) |
|---|---|---|
| A | 33.1 | 103 |
| | 34.4 | 39 |
| | 47.6 | 55 |
| B | 31.4 | 15 |
| | 33.5 | 55 |
| | 36.7 | 46 |
| | 46.7 | 113 |
| | 69.4 | 104 |
| C | 56.7 | 58 |
| D | 32.6 | 55 |
| E | 29.5 | 58 |

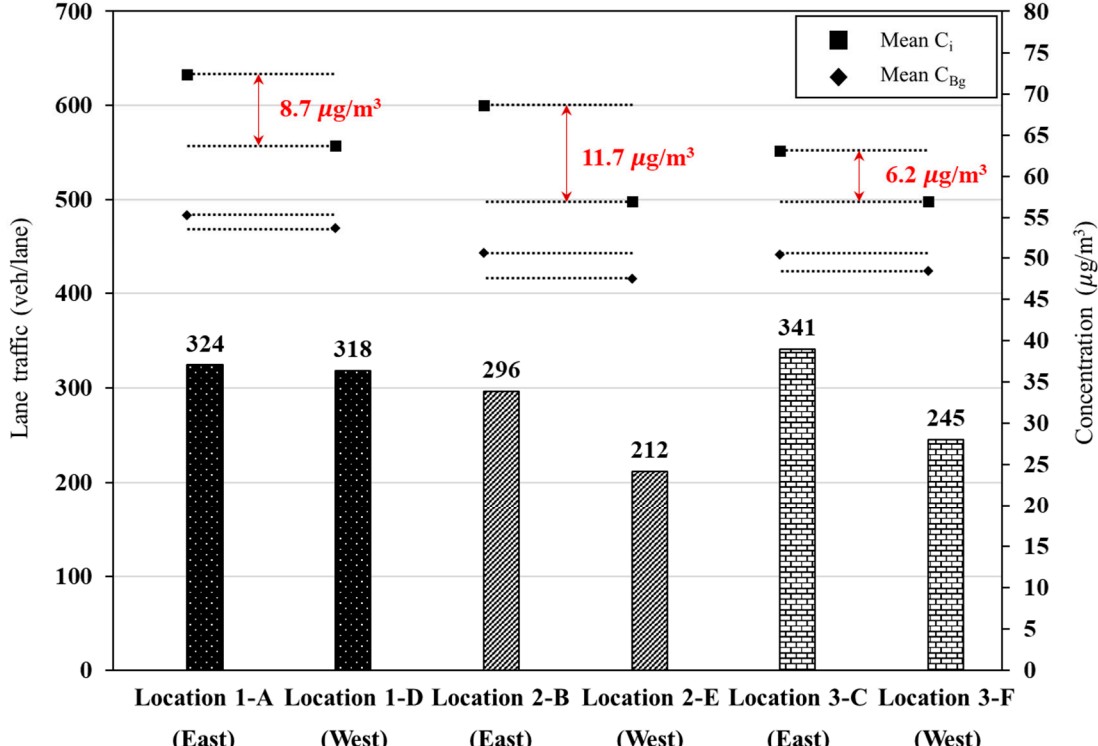

**Figure 10.** Comparison of lane traffic level and average values of $C_i$ and $C_{Bg}$ in the measurement section (the location of each measurement section was compared with Locations 1, 2, and 3, and the direction of the vehicle was the opposite side).

$C_i$ and $C_{Bg}$ showed a relatively high consistent tendency when traffic was high by location, and $C_{res}$ results when traveling in the eastern direction were higher than $C_{res}$ when traveling in the western direction, but the $C_{res}$ concentration did not appear in order depending on the traffic difference. In order to confirm the distribution of the lane traffic volume data sample, the lower quartile (25th) value or less were classified into lane traffic low level, the lower quartile (25th) value to median (50th) value were divided into lane traffic low to middle level, the median(50th) value to upper quartile (75th) value were divided into middle to high level, and the upper quartile (75th) value or higher value was divided into high level type (Figure 11).

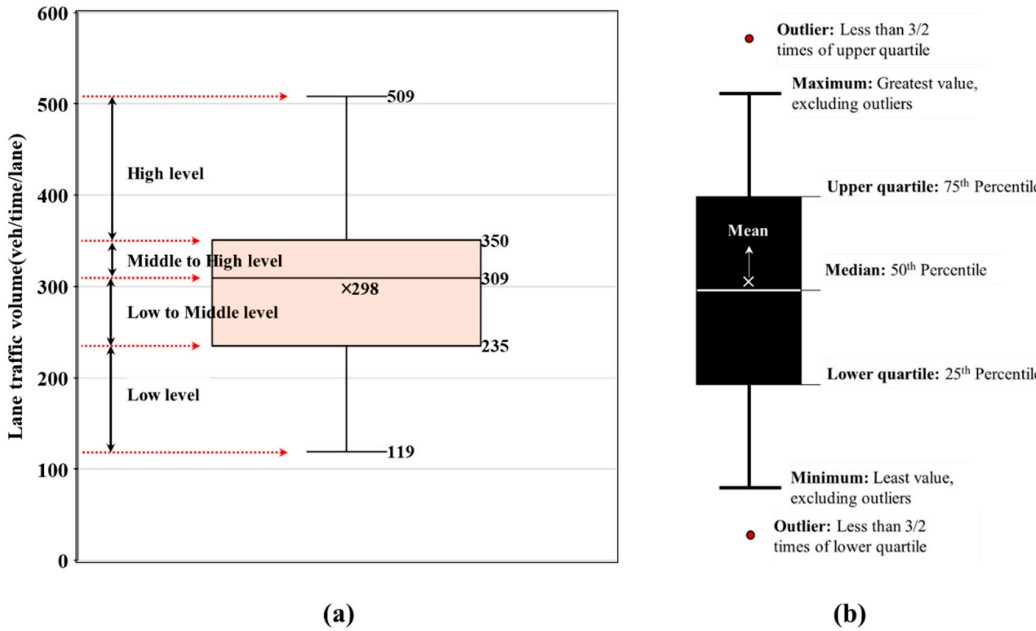

**(a)**　　　　　　　　　　　　　　　　　　　**(b)**

**Figure 11.** (**a**) Lane traffic level type classification considering lane traffic volume data distribution; (**b**) definition of boxplot used in this study.

Figure 12 shows the graph of the level type and road pollution levels $C_i$ and $C_{Bg}$ of lane traffic generated for each section. Relatively high $C_{res}$ values were observed in sections with a relatively high frequency of occurrence of high-level lane traffic by location, and the results were as follows: Location 1—Section A (11 times, $C_{res}$: 17.1 µg/m$^3$) vs. Section D (7 times, $C_{res}$: 10 µg/m$^3$); Location 2—Section B (5 times, $C_{res}$: 18 µg/m$^3$) vs. Section E (1 time, $C_{res}$: 9.4 µg/m$^3$); Location 3—Section C (16 times, $C_{res}$: 12.8 µg/m$^3$) vs. Section F (0 time, $C_{res}$: 8.5 µg/m$^3$).

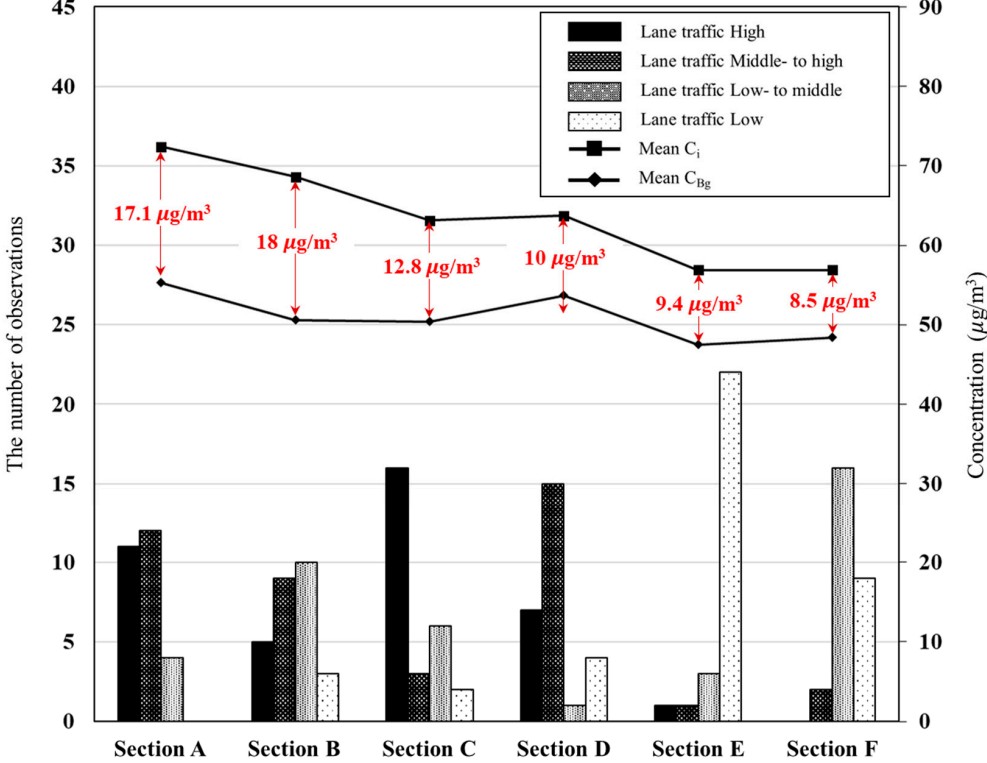

**Figure 12.** Comparison of the number of lane traffic level type observations and $C_{res}$ concentrations.

In addition, the lowest average $C_{Bg}$ (47.5 μg/m$^3$) and average $C_i$ (56.9 μg/m$^3$) concentrations were measured in Section E, where low-level lane traffic appeared 22 times.

### 3.5. Characteristics of $C_i$, $C_{Bg}$ Resuspended Dust by Lane Traffic Level by Measurement Section through Statistical Probability Comparison

As Table 5, the weather factor of Bg and the correlation coefficient R $\geq$ 0.4 were derived as the rainfall time, and a positive (+) correlation of R: 0.41 was observed. Additionally, as the visibility decreased, a negative (−) correlation of R: 0.35 was observed, $C_{Bg}$ for each observed section also derived a positive (+) correlation between the rainfall time and R: 0.35 to 0.50, and a negative (−) correlation between visibility and R: 0.33 to 0.60, and rainfall time and visibility, were shown to affect Bg and $C_{Bg}$ with similar properties.

**Table 5.** Bg, $C_{Bg}$, $C_i$ by interval and correlation analysis of major factors.

| Independent | | $C_i$ | Bg | Lane Traffic | Gr Temp. | RH | WS | Sea Level Pressure | Visibility | Rainfall Time |
|---|---|---|---|---|---|---|---|---|---|---|
| | | | | | | Correlation R-Value | | | | |
| Bg | | 0.66 | | 0.04 | −0.12 | −0.14 | −0.13 | 0.19 | −0.35 | 0.41 |
| Section A | $C_{Bg}$ | 0.95 | 0.62 | −0.21 | −0.24 | 0.36 | −0.30 | 0.35 | −0.60 | 0.44 |
| | $C_i$ | | 0.71 | −0.14 | −0.12 | 0.24 | −0.27 | 0.25 | −0.52 | 0.34 |
| Section B | $C_{Bg}$ | 0.93 | 0.67 | 0.05 | −0.28 | 0.29 | −0.29 | 0.47 | −0.59 | 0.48 |
| | $C_i$ | | 0.77 | 0.00 | −0.11 | 0.18 | −0.21 | 0.35 | −0.46 | 0.43 |
| Section C | $C_{Bg}$ | 0.96 | 0.70 | −0.09 | −0.29 | 0.16 | −0.32 | 0.38 | −0.50 | 0.44 |
| | $C_i$ | | 0.74 | −0.17 | −0.27 | 0.14 | −0.30 | 0.34 | −0.41 | 0.38 |
| Section D | $C_{Bg}$ | 0.99 | 0.49 | 0.29 | −0.14 | 0.08 | −0.36 | 0.47 | −0.40 | 0.35 |
| | $C_i$ | | 0.52 | 0.27 | −0.14 | 0.04 | −0.32 | 0.45 | −0.35 | 0.34 |
| Section E | $C_{Bg}$ | 0.98 | 0.52 | 0.53 | −0.19 | 0.26 | −0.33 | 0.35 | −0.36 | 0.49 |
| | $C_i$ | | 0.56 | 0.55 | −0.22 | 0.24 | −0.29 | 0.34 | −0.28 | 0.43 |
| Section F | $C_{Bg}$ | 0.99 | 0.69 | 0.35 | −0.31 | 0.23 | −0.30 | 0.48 | −0.56 | 0.47 |
| | $C_i$ | | 0.70 | 0.34 | −0.30 | 0.24 | −0.27 | 0.47 | −0.53 | 0.46 |

This was similar to the results of a study that showed that the fine dust on the road increased over time after rainfall [39]. Additionally, the correlation coefficient between $C_{Bg}$ and other weather factors was R: −0.13 to −0.31 in Gr temp., R: −0.16 to −0.36 in WS, R: 0.27 to 0.48 in sea level pressure, and the correlation coefficient between Bg and factors was higher than the Gr temp. (R: −0.12), WS (R: −0.13) and sea level pressure (R: 0.19).

As a result of analyzing the correlation between $C_{Bg}$ and $C_i$ concentrations and meteorological factors by section, the correlation R-value with $C_i$ was consistently lower than the $C_{Bg}$ correlation R-value and major weather factors (sea level pressure, visibility, and rainfall time) with R $\geq$ 0.4. However, a consistent trend was not observed in the correlation R-value with $C_i$ and $C_{Bg}$ concentrations according to the change of lane traffic by section.

We tried to analyze the characteristics of resuspended PM$_{10}$ concentration according to the lane traffic level type by classifying lane traffic into level types and comparing the observed atmospheric weather environment level and resuspended PM$_{10}$ concentration.

Table 6 shows the average, 75th, median, and 25th values classified to compare $C_{res}$ and Bg and the distribution of weather factor data according to the lane traffic level type.

**Table 6.** Distribution of 25th, median, mean, and 75th percentile data by lane traffic level factors.

| Lane Traffic Level | Percentile of Data | $C_{res}$ (µg/m³) | Bg (µg/m³) | Gr Temp (°C) | RH (%) | WS (m/s) | Sea Level Pressure (hPa) | Visibility (m) | Rainfall Time (Days) |
|---|---|---|---|---|---|---|---|---|---|
| High (over 350 veh/time/lane) | 75th | 14.0 | 93.0 | 19.6 | 54.0 | 4.0 | 1025.9 | 1996.3 | 7.0 |
| | Mean | 11.9 | 53.2 | 15.9 | 48.1 | 3.0 | 1022.1 | 1685.2 | 5.4 |
| | Median | 9.6 | 50.0 | 13.3 | 49.5 | 3.1 | 1020.6 | 1932.0 | 4.0 |
| | 25th | 6.1 | 23.5 | 11.5 | 36.8 | 1.6 | 1019.5 | 1635.0 | 3.0 |
| Middle to high (309 to 350 veh/time/lane) | 75th | 16.5 | 93.0 | 26.0 | 68.0 | 3.7 | 1021.6 | 1995.0 | 7.0 |
| | Mean | 12.6 | 57.8 | 21.0 | 47.5 | 2.6 | 1019.6 | 1703.5 | 5.9 |
| | Median | 11.1 | 56.0 | 21.4 | 44.0 | 2.5 | 1019.5 | 1937.0 | 6.0 |
| | 25th | 7.2 | 34.0 | 13.3 | 31.0 | 1.5 | 1018.0 | 1527 | 4.0 |
| Low to middle (235 to 309 veh/time/lane) | 75th | 20.1 | 70.0 | 24.8 | 53.0 | 3.8 | 1025.6 | 1995.0 | 6.0 |
| | Mean | 15.3 | 54.0 | 19.0 | 46.1 | 2.8 | 1021.1 | 1736.0 | 4.7 |
| | Median | 10.7 | 51.0 | 19.6 | 46.0 | 2.8 | 1020.5 | 1932.0 | 4.0 |
| | 25th | 8.0 | 34.0 | 13.3 | 31.0 | 1.6 | 1018.8 | 1685.0 | 4.0 |
| Low (under 235 veh/time/lane) | 75th | 11.6 | 61.0 | 25.5 | 68.3 | 3.8 | 1024.8 | 1986.8 | 6.0 |
| | Mean | 10.1 | 54.4 | 17.0 | 53.0 | 2.6 | 1021.0 | 1588.0 | 4.8 |
| | Median | 8.9 | 51.0 | 18.7 | 52.0 | 2.5 | 1020.6 | 1895.5 | 5.0 |
| | 25th | 8.3 | 35.5 | 9.8 | 35.0 | 1.6 | 1018.0 | 913 | 3.8 |

Among the data samples classified by the lane traffic level type, the order in which the $C_{res}$ mean observed was found to be large was lane traffic low to middle level (15.3 µg/m³) > lane traffic middle to high level (12.6 µg/m³) > lane traffic high level (11.9 µg/m³) > lane traffic low level (10.1 µg/m³). Results measured at the lane traffic high-level Bg were mean: 53.2 µg/m³; median: 50.0 µg/m³; 25th to 75th value: 23.5 µg/m³ to 93.0 µg/m³; then, the Bg of the good level (under the results measured at the broadest range of Bg levels, from 31 µg/m³) to the bad level (81 to 151 µg/m³). Relatively high WS, sea level pressure, rainfall time 75th value (7.0 days), and high visibility 25th value (1635 m) were measured.

Results measured at the lane traffic middle to high level Bg were: mean: 57.8 µg/m³; median: 56.0 µg/m³; 25th to 75th value: 34.0 µg/m³ to 93.0 µg/m³; highest Bg concentration and Gr temp; rainfall time values were measured.

Results measured at the lane traffic low to middle level Bg were: mean: 54.0 µg/m³; median: 51.0 µg/m³; 25th to 75th value: 34.0 µg/m³ to 70.0 µg/m³; relatively high WS (1.6 to 3.8 m/s); sea level pressure (1018.8 to 1025.6 hPa); visibility (1685 to 1995 m) in a wide range.

Results measured at the lane traffic low level at the Bg level in the narrowest range were: mean: 54.4 µg/m³; median: 51.0 µg/m³; 25th to 75th values: 35.5 µg/m³ to 61.0 µg/m³. RH had a relatively high 75th value (68.3%); the visibility 25th value (913 m) was the lowest.

Figure 13 shows a box plot that compares the $C_{Bg}$ and $C_i$ mean values and the 25th, median, and 75th values calculated by the lane traffic level type, comparing the distribution of data samples, and analyzing the characteristics of road resuspended dust concentrations measured by traffic class. The maximum 25th value of $C_{Bg}$ was 19.7 µg/m³ (lane traffic high level), and the minimum value was 18.1 µg/m³ (lane traffic low to middle level). The maximum median value of $C_{Bg}$ was 47.8 µg/m³ (lane traffic low level), and the minimum value was 32.7 µg/m³ (lane traffic low to middle level). The maximum 75th value of $C_{Bg}$ was 75.3 µg/m³ (lane traffic high Level), and the minimum value was 67.1 µg/m³ (lane traffic low Level). The maximum 25th% value of $C_i$ was 36.6 µg/m³ (lane traffic low to Middle level), and the minimum value was 30.4 µg/m³ (lane traffic low level), The maximum median value of $C_i$ was 63.9 µg/m³ (lane traffic low Level), and the minimum value was 54.5 µg/m³ (lane traffic low to middle Level). The maximum 75th value was 85.1 µg/m³ (lane traffic high level), and the minimum value was 77.7 µg/m³ (lane traffic low level).

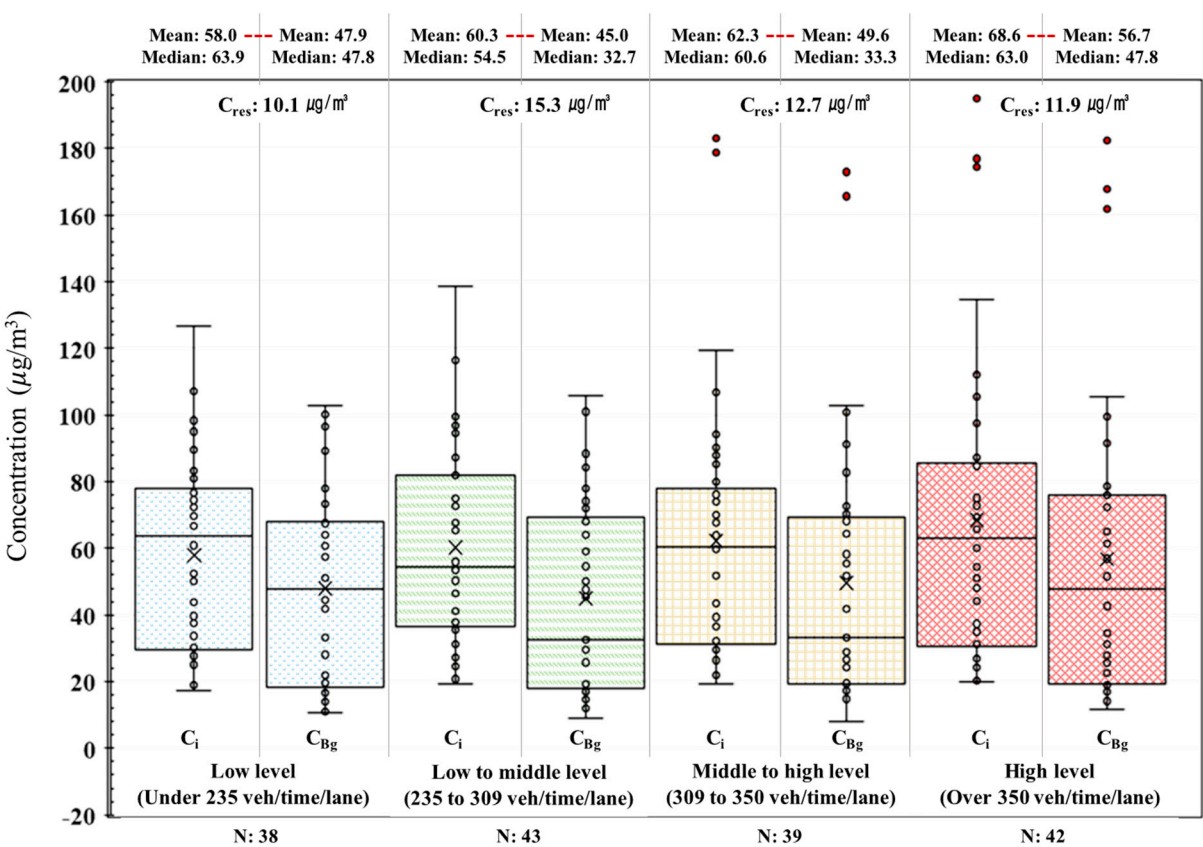

**Figure 13.** Comparison of Box plot by lane traffic level types.

The average level of lane traffic level for $C_{Bg}$ was observed to be relatively high in the order of lane traffic low to middle level (45.0 µg/m³) < lane traffic low level (47.9 µg/m³) < lane traffic middle to high level (49.6 µg/m³) < lane traffic high level (56.7 µg/m³). The average level of $C_i$ was observed in the order of lane traffic low level (58.0 µg/m³) < lane traffic low to middle level (60.3 µg/m³) < lane traffic middle to high level (62.3 µg/m³) < lane traffic high level (68.6 µg/m³). Compared to over the middle-level lane traffic, under middle-level lane traffic produced an average small difference of Bg and observed high $C_{Bg}$ and $C_i$ levels, resulting in low $C_i$ and $C_{Bg}$ for relatively low volumes of traffic and high $C_i$ and $C_{Bg}$ for high volumes of traffic. The overall pollution level of the road may have increased as the lane traffic level increased. However, when the traffic volume increased, the relative road atmospheric pollution also increased, so the $C_{res}$ level did not differ significantly.

## 4. Conclusions

The main goal in this paper was to quantify the characteristics of weather and traffic conditions on $PM_{10}$ concentrations measured using the resuspended $PM_{10}$ measurements system and to identify the main characteristics of $PM_{10}$ concentrations. Technical statistics, a correlation analysis, and a comparison and analysis were carried out according to changes in the traffic levels.

The four roads, which varied by region with limited traffic, were easy sections to observe the concentration of resuspended dust ($C_{res}$) increased by speed and dust load (sL) on the road. In addition, the six roads where traffic was observed were composed of auxiliary highways in urban areas, where it was easy to collect traffic information at the same time as conducting air pollution monitoring of adjacent stations and have conditions to grasp the effect of $PM_{10}$ concentration. The following are the results identifying the characteristics of sL measured on roads with limited traffic, the $PM_{10}$ concentration of resuspended dust, and $PM_{10}$ measured in six sections of the urban area during traffic operation.

$C_i$ increased with increasing speed from 30 km/h to 60 km/h, and the highest increase was when the speed increased from 30 km/h to 40 km/h. At this time, $C_{Bg}$ showed little change in the same measured environment. Therefore, the dispersion of the resuspended dust measured was expected to have an effect on the sL in the section due to the wind generated by the speed of driving.

On different measured roads, higher sL was observed compared with urban roads operating in areas with a clear external dust inflow, such as roads in construction complexes. In addition to the empty site, the sL collected on the same road was observed to increase over time without an adjustment of the external environment and traffic intervention.

The sL measured on paved roads in the testbed (restricted access to traffic) and on the construction site was observed to be 0.034 g/m² to 0.093 g/m². In the case of 0.034 g/m² with sL, the concentration of resuspended dust was found to be in the range of 47 μg/m³~225 μg/m³. Compared to this value, an sL distribution below 0.009 g/m² could produce from 11 μg/m³ up to 29 μg/m³ of resuspended dust, which was judged to be lower than 0.034 g/m² of sL. Therefore, with a higher level of sL, one can generate a high level of road resuspended dust.

Even if sL could produce effective road resuspended dust, the background concentration ($C_{Bg}$) during driving was not affected by the surrounding area. Although it had almost the same level as the concentration of adjacent stations (Bg) ($R^2 = 0.84$), it had a low correlation with the concentration of fine dust floating due to it being suspended continuously on roads where traffic operated. In the event of yellow dust, the difference between the background concentration level and the air pollution level on the road was large, indicating a negative correlation ($R^2 = 0.10$).

Using the resuspended dust $PM_{10}$ monitoring results for six sections of roads under traffic operation during the same period, the correlation analysis results between traffic and atmospheric weather environmental factors were as follows:

-   A significant correlation coefficient between Bg and R > 0.3 was derived from time after rain (R = 0.41) and visibility (R = −0.35), $C_{Bg}$ by section was time after rain (R = 0.35 to 0.50) and visibility (R = −0.60 to −0.33), and time after rain and visibility were found to have similar characteristics to Bg and $C_{Bg}$.

As a result of analyzing the correlation between $C_{Bg}$ and $C_i$ concentrations and meteorological factors by section, the correlation R-value with $C_i$ was consistently lower than the $C_{Bg}$ correlation R-value and major weather factors (sea level pressure, visibility, and rainfall time) with R $\geq$ 0.4. However, a consistent trend was not observed in the correlation R-value with $C_i$ and $C_{Bg}$ concentrations according to the change of lane traffic by section.

The results of resuspended dust $PM_{10}$ monitoring of six road sections operating during the same period were compared and analyzed by location, direction, and lane traffic level types in the measurement areas as follows:

-   A lane traffic level type for whole data was compared by dividing the upper distribution of the lower quartile (25th) value into a high-level type, median to 75th value distribution grade into middle to high-level type, and the 25th to median value range distribution grade into low-level type.
-   Higher $C_{res}$ values were observed consistently in sections with a relatively high frequency of occurrence of high levels of lane traffic among sections traveling in the same direction on roads divided by location.
-   The lowest average $C_{Bg}$ (45.0 μg/m³) and average $C_i$ (56.7 μg/m³) concentrations were measured in Section E, where low levels of lane traffic were observed the most, i.e., 22 times.
-   Among the data samples classified by lane traffic level type, the order in which the $C_{res}$ mean was observed and found to be large was lane traffic low to middle level (15.3 μg/m³) > lane traffic middle to high level (12.7 μg/m³) > lane traffic high level (11.9 μg/m³) > lane traffic low level (10.1 μg/m³).

- The average level of lane traffic level $C_{Bg}$ was observed to be relatively high in the order of lane traffic low to middle level (45.0 µg/m$^3$) < lane traffic low level (47.9 µg/m$^3$) < lane traffic middle to high level (49.6 µg/m$^3$) < lane traffic high level (56.7 µg/m$^3$). The average level of $C_i$ was observed in the order of lane traffic low level (47.9 µg/m$^3$) < lane traffic low to middle level (49.6 µg/m$^3$) < lane traffic middle to high level (62.3 µg/m$^3$) < lane traffic high level (68.6 µg/m$^3$).
- The overall pollution level of the road may have increased as lane traffic level increased. However, when the traffic volume increased, the relative road atmospheric pollution also increased, so the $C_{res}$ level did not differ significantly.

The characteristics of the difference between the concentration of resuspended dust ($C_i$) and the background concentration of roads ($C_{Bg}$) and the background of city atmosphere (Bg) concentration measured were compared with the effects of traffic and weather conditions. In the case of Korea, PM reduction measures are being implemented according to the occurrence of high concentrations of PM$_{10}$ and PM$_{2.5}$ provided by the city Bg observations. However, results of this study suggested the need for an efficient alternative considering the effect of yellow dust over time, because due to the occurrence of yellow dust, Bg can differ to the resuspended dust concentration. Additionally, the concentration of resuspended dust on roads may differ significantly from that of the adjacent Bg observation caused by vehicles driving on roads. Therefore, it is suggested that a more frequent occurrence of high levels of resuspended dust may occur, and research should be continued to quantify the influence of dust collected through mobile measurements and to provide accurate forecasts.

**Author Contributions:** Conceptualization, I.K. and S.H.; methodology, H.Y. and G.Y.; investigation, J.C. and H.Y.; original draft preparation, S.H.; writing—review and edition, I.K. and G.Y. All authors have read and agreed to the published version of the manuscript.

**Funding:** This work was supported by the Korea Agency for Infrastructure Technology Advancement (KAIA) grant funded by the Ministry of Land, Infrastructure, and Transport (grant number 21POQWB152342-03) and the National Research Foundation of Korea (grant number NRF-2019R1A2C1007224).

**Institutional Review Board Statement:** Not applicable.

**Informed Consent Statement:** Not applicable.

**Data Availability Statement:** The urban weather observation Automated Synoptic Observing System(ASOS) data provided by Korea Meteorological Administration (KMA). Available online: https://data.kma.go.kr, accessed on 27 July 2022; The road traffic observation Vehicle Detect System(VDS) data provided by Yong-In city in Korea. Available online: http://its.yongin.go.kr/trafficStats/road.do, accessed on 27 July 2022.

**Acknowledgments:** The authors would like to thank the members of the research team, KAIA, and MOLIT for their guidance and support throughout the project.

**Conflicts of Interest:** The authors declare no conflict of interest.

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
