# Peer review of "Characteristics of Resuspended Road Dust with Traffic and Atmospheric Environment in South Korea"

_atmosphere, doi:10.3390/atmos13081215_

Round 1

Reviewer 1 Report

The revised manuscript was added most of the requested information, which including the informations need to be presented in Abstract, important technology parameters for Methods. The raised questions also were explained clearly. The grammar of the manuscript had been improved and it is appropriate for publishing. Overall, I recommend acceptance of the manuscript. 

Author Response

Appreciate for taking a close review at our Manuscript.

We are submitting a word file containing your responded to the of comments by the reviewers.

We have carefully responded to the reviewers' detailed reviews.

We heartily thank you.

Reviewer 2 Report

Please see below for the comments (minor modifications):

-          Line 82: It is better to reference them as “meteorological variables” rather than “weather”

-          Line 88-89: Rephrase and rewrite the sentence

-          Line 90: “Background” is one word

-          Improve the quality of Fig 1

-          Add lines to Table 1 to clearly show the rows related to the two categories in column 1.

-          Line 117-118: Correct the ref.

-          Line 120-121: Explain the difference between suspended and resuspended dust, and use it throughout the article.

-          Please read these two references (for their methodology) and reference them in your article:

o   Near-Road Traffic-Related Air Pollution: Resuspended PM2.5 from Highways and Arterials

o   Traffic contribution to PM2.5 increment in the near-road environment

-          Modify all cases you have a comma at the end of a sentence (instead of a period). Line 155 is an example.

-          Line 165: What does ash suspended mean?!

-          Line 173: Please remove “During the monitoring period”

-          Line 175-178: You can use () instead of many commas in one sentence.

-          Line 178: The word “judge’ can be replaced with other words

-          What does kph-1 mean? Is it km/h? Please correct all instances thoroughly

-          You can use a table for lines 206-208, and 318-321.

-          There is no need to have a period after the number of tables or figures when they are not at the end of a sentence. Line 216 is an instance (and line 402)

-          Please use the same font in figures and tables. Figure 3 is an example.

-          Line 447-451: break into two sentences.

-          Lines 453-454: rewrite it

-          In the Conclusion section, please summarize items 1 to 6 in one paragraph, and explain what we learn from this data collection study, very high-level.

Author Response

Appreciate for taking a close review at our Manuscript.

We submit a word file written in response to the reviewers' comments.

We have carefully responded to the reviewers' detailed reviews.

We heartily thank you.

Author Response

(The authors gave the same response as above.)
